# COVID-19 and children with congenital anomalies: a European survey of parents' experiences of healthcare services

Anna Latos-Bieleńska,[1] Elena Marcus [iD],[2] Anna Jamry-Dziurla,[1] Judith Rankin,[3] Ingeborg Barisic,[4] Clara Cavero-Carbonell [iD],[5] Elly Den Hond,[6] Ester Garne [iD],[7] Lucas Genard,[6] Ana João Santos,[8] L Renée Lutke,[9] Carlos Matias Dias,[10] Christina Neergaard Pedersen,[7] Amanda Neville,[11] Annika Niemann,[12] Ljubica Odak,[4] Lucía Páramo-Rodríguez [iD],[5] Anna Pierini,[13] Anke Rissmann [iD],[12] Joan K Morris [iD] [14]

AL-Bńs and EM contributed equally.

AL-Bńs and EM are joint first authors.

**Correspondence to**
Dr Elena Marcus;
emarcus@sgul.ac.uk

## ABSTRACT

**Objective** To survey parents and carers of children with a congenital anomaly across Europe about their experiences of healthcare services and support during the COVID-19 pandemic.

**Design** Cross-sectional study.

**Setting** Online survey in 10 European countries, open from 8 March 2021 to 14 July 2021.

**Population** 1070 parents and carers of children aged 0–10 years with a cleft lip, spina bifida, congenital heart defect (CHD) requiring surgery and/or Down syndrome.

**Main outcome measures** Parental views about: the provision of care for their child (cancellation/postponement of appointments, virtual appointments, access to medication), the impact of disruptions to healthcare on their child's health and well-being, and satisfaction with support from medical sources, organisations and close relationships.

**Results** Disruptions to healthcare appointments were significantly higher (p<0.001) in the UK and Poland, with approximately two-thirds of participants reporting 'cancelled or postponed' tests (67/101; 256/389) and procedures compared with approximately 20% in Germany (13/74) and Belgium/Netherlands (11/55). A third of participants in the UK and Poland reported 'cancelled or postponed' surgeries (22/72; 98/266) compared with only 8% in Germany (5/64). In Poland, 43% (136/314) of parents reported that changes to their child's ongoing treatment had moderately to severely affected their child's health, significantly higher than all other countries (p<0.001). Satisfaction ratings for support from general practitioners were lowest in the UK and Poland, and lowest in Poland and Italy for specialist doctors and nurses.

**Conclusion** A large proportion of participants reported disruptions to healthcare during the pandemic, which for some had a significant impact on their child's health. Regional differences in disruptions raise questions about the competence of certain healthcare systems to meet the needs of this vulnerable group of patients and indicate improvements should be strived for in some regions.

## STRENGTHS AND LIMITATIONS OF THIS STUDY

⇒ Surveys of the experiences of a large total number of parents and carers across several European countries and congenital anomaly (CA) types. The proportion of each CA type in the study sample reflects the relative number of live births with each CA in Europe.

⇒ High item-level response rates suggest that survey items were relevant to participants and easy to complete.

⇒ Potential bias in responses due to the use of social media for recruitment, for example, excluding people living with 'digital poverty' and those who do not engage with patient and parent organisations, limiting the generalisability of findings.

⇒ Inability to conduct a full pilot of the final survey to explore item acceptability, comprehension and relevance, possible that there may be some issues with the wording or content of items.

## BACKGROUND

The COVID-19 pandemic put pressure on healthcare systems worldwide, causing severe disruptions to the delivery of non-essential services, as staff were redeployed to acute care, and outpatient treatment and follow-up were reduced due to concerns about viral transmission in hospital.[1–3] Non-urgent elective care was the most heavily impacted, with a record backlog of 5.6 million cases reported in England in July 2021.[4 5]

Congenital anomalies (CAs) are a range of conditions that are present from birth and remain a leading cause of childhood morbidity and long-term disability.[6 7] Children with CAs require regular clinical follow-up,[8] including more frequent primary care appointments, hospital admissions and surgeries than

children without CAs.[9–12] Although children are less affected by SARS-CoV-2 infection than adults,[13] Down syndrome has been indicated as a risk factor of severe disease and mortality,[14 15] and children with underlying conditions may be at increased risk of infection.[16] It is crucial to document the healthcare experiences of children with CAs during this period of increased pressure on healthcare systems, especially as they represent a vulnerable population. Existing research, conducted during the first wave of the pandemic in 2020, suggests a high proportion of cancellations and postponements to paediatric healthcare appointments and treatments in the USA[17] and in Europe.[18–22] Disruptions to the healthcare services of children with CAs were found to cause anxiety for parents,[20] and fear that their child's health may be negatively affected.[22] Corcerns about SARS-CoV-2 infection were also common among parents,[23 24] which coupled with reductions in other communicable infections during the pandemic,[25 26] resulted in fewer visits to clinics[25 27] and emergency departments[26 28] in 2020. Parents reported a lack of support from healthcare professionals, including the absence of specific COVID-19-related guidance for children.[22 29]

This paper describes a cross-sectional online survey, which explored the views of parents and carers of children with CAs about: (a) their healthcare experiences and (b) their experiences of support, 1 year into the pandemic. The survey was conducted as part of a collaborative European project, 'Establishing a linked European Cohort of Children with CAs (EUROlinkCAT)',[30] which aims to investigate health and educational outcomes in children born with CAs using population-based data. Due to differences in the level of restrictions, healthcare systems and the availability of resources between countries, the survey was conducted in several European countries, to explore possible variations in the provision of care.

## METHODS
This study is reported following the Strengthening the Reporting of Observational studies in Epidemiology guidelines.[31] The findings presented are a subsection of a cross-sectional online survey, conducted by the EUROlinkCAT team, which explored the wider information and support needs of parents and carers of children with CAs in 10 European countries. This paper focuses on the healthcare experiences and health status of children during the COVID-19 pandemic and parent and carer experiences of support. The survey was launched in the UK and Poland on 8 March 2021 and kept open until 14 July 2021. The survey was launched in a staggered manner in each country, as and when translations were finalised and approvals granted (table 1).

### Participants
The survey was open to parents, carers and guardians (termed, henceforth, as *parents*) of children up to 10 years of age who have one or more of the following CAs: cleft

lip (with or without a cleft palate), spina bifida, congenital heart defect (CHD) which required surgery and Down syndrome. Due to the high level of heterogeneity across all CAs, these groups were predefined and selected to cover different types of impairments, with likely differing impacts on the experiences of the child and parent: (a) physical disability (spina bifida), (b) learning disability (Down syndrome), (c) visible defects (cleft lip) and (d) non-visible defects (CHD). Participants were actively recruited in 10 European countries: Belgium, Croatia, Denmark, Germany, Italy, Netherlands, Poland, Portugal, Spain, and the UK.

### Recruitment
Participants were recruited with convenience sampling which was conducted online via social media (Twitter and Facebook), charities and patient organisations within each participating country (eg, the Down Syndrome Association in the UK) and closed support groups on Facebook. Potential participants were provided with a link to the survey website, which included all language versions of the survey. Participants were provided with the participant information sheet at the start of the survey, and depending on local ethics requirements, participants were either required to complete an online consent form or consent was implied by completion of the survey. As the survey was shared across online platforms and by a number of international organisations (eg, Down Syndrome International), responses were also received from parents living in other European countries (eg, Ireland), and these were retained in the analysis.

### Survey
The content of the survey was developed following a literature review, and input from expert clinicians, parents and educators, academics with expertise in CA research and questionnaire development and a Public Involvement and Community Engagement lead. The survey included the following sections: (1) Parent Demographics (nine items), (2) Child Demographics and Medical Information (seven items), (3) Provision of Healthcare (seven items), (4) Impact on the Child (three items) and (5) Support for Parents (two items) (see online supplemental file 1). Response options varied and comprised: yes and no (provision of healthcare); not at all, a little, quite a bit and very much (provision of healthcare); not at all satisfied, a little satisfied, quite satisfied and very satisfied (support for parents); and much worse, somewhat worse, about the same, somewhat better, much better (impact on child). All items were close ended; therefore, quantitative data only were collected. In relation to the timeframe, participants were asked to reflect on their experiences from the start of the pandemic in January 2020 to the time at which they completed the survey (March–July 2021).

### Translation
The survey was developed in English and translated into eight European languages following existing guidance.[32]

**Table 1** Recruitment period and participant characteristics by country group

| Characteristic | All | UK | Poland | Germany | Croatia | Italy | Belgium/Netherlands | Other EU* |
|---|---|---|---|---|---|---|---|---|
| **Recruitment period†** | | | | | | | | |
| Start date | – | 8 March 2021 | 8 March 2021 | 11 May 2021 | 26 April 2021 | 16 June 2021 | 19 April 2021 | 6 April 2021 |
| End date | – | 14 July 2021 | 14 July 2021 | 14 July 2021 | 14 July 2021 | 31 July 2021 | 14 July 2021 | 14 July 2021 |
| N | 986 | 120 | 476 | 97 | 68 | 59 | 74 | 92 |
| **Age** | | | | | | | | |
| ≤30 | 162 (17%) | 18 (15%) | 93 (20%) | 13 (13%) | 8 (12%) | 4 (7%) | 15 (20%) | 11 (12%) |
| 31–40 | 516 (53%) | 53 (45%) | 264 (56%) | 51 (53%) | 37 (55%) | 27 (46%) | 35 (47%) | 49 (53%) |
| >40 | 301 (31%) | 47 (40%) | 115 (24%) | 33 (34%) | 22 (33%) | 28 (47%) | 24 (32%) | 34 (35%) |
| **Relation to child** | | | | | | | | |
| Mother | 911 (92%) | 116 (97%) | 449 (94%) | 81 (84%) | 63 (93%) | 52 (88%) | 64 (86%) | 86 (95%) |
| Father | 65 (7%) | 2 (2%) | 24 (5%) | 13 (13%) | 5 (7%) | 6 (10%) | 10 (14%) | 5 (5%) |
| Other‡ | 8 (1%) | 1 (1%) | 3 (1%) | 3 (3%) | – | 1 (2%) | – | – |
| **Employment** | | | | | | | | |
| Employed | 586 (60%) | 81 (68%) | 223 (47%) | 61 (62%) | 54 (79%) | 44 (75%) | 61 (82%) | 62 (69%) |
| Homemaker/carer | 301 (31%) | 36 (30%) | 198 (42%) | 27 (29%) | 7 (10%) | 11 (19%) | 8 (11%) | 14 (16%) |
| Other§ | 94 (9%) | 3 (3%) | 52 (11%) | 9 (9%) | 7 (10%) | 4 (7%) | 5 (7%) | 14 (16%) |
| **Education** | | | | | | | | |
| School≤18 years | 390 (40%) | 44 (37%) | 163 (35%) | 61 (67%) | 19 (28%) | 30 (52%) | 44 (60%) | 29 (32%) |
| University | 482 (49%) | 50 (42%) | 257 (53%) | 27 (29%) | 45 (66%) | 19 (33%) | 29 (39%) | 55 (60%) |
| Post-graduate | 106 (11%) | 25 (21%) | 56 (11%) | 3 (3%) | 4 (6%) | 9 (16%) | 1 (1%) | 8 (9%) |
| **Migrant status** | | | | | | | | |
| >10 years/from birth | 924 (94%) | 111 (93%) | 467 (98%) | 86 (88%) | 64 (94%) | 50 (86%) | 71 (96%) | 75 (81%) |
| 6–10 years | 30 (3%) | 5 (4%) | 5 (1%) | 6 (7%) | 2 (3%) | 4 (7%) | 1 (1%) | 7 (8%) |
| 1–5 years | 28 (3%) | 4 (3%) | 2 (0.4%) | 5 (5%) | 2 (3%) | 4 (7%) | 2 (3%) | 9 (10%) |
| <1 year | 2 (0.2%) | – | 1 (0.2%) | – | – | – | – | 1 (1%) |

*Other European countries: Denmark (n=39), Portugal (n=23), Spain (n=16), Ireland (n=5), Bulgaria (n=2), Albania (n=1), Cyprus (n=1), Lithuania (n=1), Norway (n=1), Romania (n=1), Sweden (n=1), Ukraine (n=1).
†The recall period for the survey items was from January 2020 until the time at which participants were recruited.
‡Other family member (n=3), legal guardian related to the child (n=2), legal guardian unrelated to the child (n=3).
§Unemployed (n=56), long-term sick/disabled (n=17), student (n=8), on furlough (n=12), retired (n=1).

The Dutch version was used in Belgium and the Netherlands. The survey was initially translated into Polish and Italian to check for any translatability issues, and relevant amendments were subsequently made to the English version accordingly. These languages were selected because they have different origins (Slavic and Romance) with differing translation issues, and the research team included native Polish (AL-B, AJ-D) and native English-Italian bilingual (EM) speakers. Translations were carried out in four steps for each language version: (1) a native speaker of the target language with good command of English conducted the initial translation, (2) the translation was checked by at least one other native speaker of the target language and any problems discussed and reconciled, (3) the survey was back-translated by a native English speaker (or person with a good command of the English language) and who was naïve to the original version, (4) the back-translated survey was reviewed by EM against the original English language version and any semantic or conceptual discrepancies in the back-translation were flagged and discussed with the translators until they were resolved. Due to differences in education systems across Europe, equivalent terminology for participants' education level was not available, and categories were selected to reflect local education systems within each country.

### Data collection

Study data were collected and managed using Research Electronic Data Capture[33] tools hosted at St George's, University of London. All data collected were anonymous and it was, therefore, not possible to verify CA diagnoses. To keep the survey fully anonymous, no internet protocol addresses were collected, so it was not possible to prevent multiple participation. Participants were initially allowed to skip any item, however, following an interim analysis on 27 April 2021, a high proportion of missing data for country and CA type was noted. As these data were crucial to the research question, these two items were subsequently made mandatory.

### Patient and public involvement

People with experience in caring for or teaching children with CAs contributed to the development of the survey. These were (a) three parents of children with CAs who also run patient organisations/charities relevant to their child's condition (Down syndrome, spina bifida, metachromatic leukodystrophy), (b) a clinical geneticist who works closely with a number of parent organisations and (c) a teacher of children with special educational needs and disabilities. Each individual commented on an early draft of the survey, including the overall content of the survey and the wording of questions and response options. This feedback was reviewed by the research team and relevant modifications were made to the draft survey to address it.

Findings from the study will be shared with members of the public, parents and carers, healthcare professionals and relevant stakeholders via scientific publications, lay reports, social media and conferences.

### Data analysis

Descriptive statistics were conducted using Stata V.17.0 software.[34] Data were checked to ensure that answers were consistent (eg, identifying if a mother replying that she is 20–25 years old is retired). Outcomes scored on 4-point Likert scales were dichotomised (very much/satisfied vs other responses) and outcomes scored on a 5-point Likert scale were collapsed into three categories by merging the two lowest response options (much worse and somewhat worse) and the two highest response options (much better and somewhat better). Data were modelled using multivariate logistic regressions and ordinal logistic regressions, which included the child's anomaly type, parent's country of residence and age and education level. The impact of country and anomaly type on outcomes was explored, choosing the largest categories as the comparator groups (Poland and CHD). For age and education, categorical data were collected. For the analysis, each variable was recoded into three groups: age (<30 years; 31–40 years, >40 years), education (formal education until 16 or 18 years/technical training; university degree; postgraduate degree). Age and education were included in our regression models as ordinal variables. To control for multiple comparisons, the alpha level was adjusted to p<0.01 for all analyses. It was unlikely that data were missing at random, so more sophisticated multiple imputation techniques were not adopted.

We aimed to recruit 80 participants per country which would have resulted in a power of 80% to determine that a country with 20% of participants replying category 4 (very much/satisfied) was statistically significantly different at the 95% level of significance from a country with 40% of participants replying category 4. Owing to delays in obtaining ethics approvals, this target was not met within the timescales for some countries. Data were presented by country if these were available for at least 50 participants. Where there were <50 participants, data were combined into an 'other European country' group (termed, henceforth, as *Other EU*), which included participants from a heterogeneous group of countries. Due to similarities in survey responses, geographical location and language, data for Belgium (n=46) and the Netherlands (n=28) were combined into a single group. For CAs, data were categorised according to the four anomalies, and a separate category was created for children with Down syndrome and a CHD, a common comorbidity.[35] There were too few participants to create meaningful categories for children who had other combinations of the four anomalies (n=15), and these were excluded from the analysis.

Given that a multimodal online recruitment strategy was used, it was not possible to estimate how many potential participants the survey reached in order to calculate response rates.[36] We report submission rates (number of participants who started the survey/number who completed

  Latos-Bieleńska A, *et al. BMJ Open* 2022;**12**:e061428. doi:10.1136/bmjopen-2022-061428

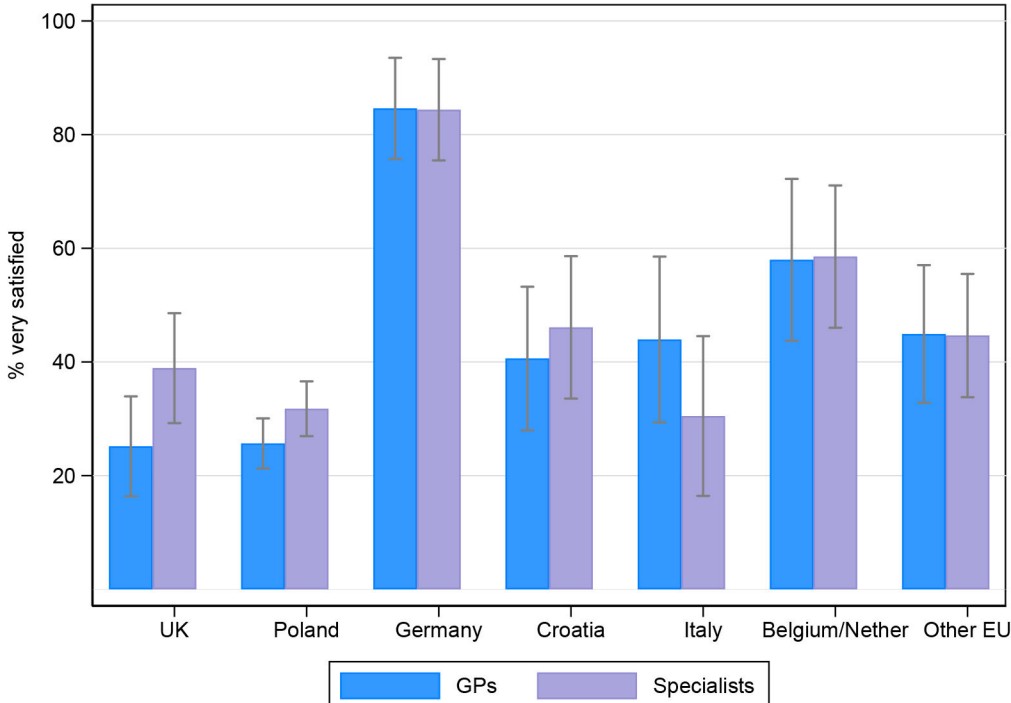

**Figure 1** Proportion* of participants reporting 'cancelled or postponed' routine appointments, planned tests or procedures, and planned surgeries with 95% CIs, by country. *Adjusted by congenital anomaly type, parental age and education level.

and submitted the survey).[37] For those who submitted their survey, we report item-level response rates (proportion of participants completing each item)[38].

## RESULTS
### Participant characteristics
1298 parents across Europe accessed the survey, of whom 1109 (85%) submitted their responses. The submission rate ranged from 78% in Italy to 92% in Germany and Belgium. A further 123 (9.5%) submitted forms were not included in the analysis as country data were missing (n=80), CA data were missing (n=24), participants were from non-European countries (n=4) or participants specified different combinations of the four anomaly types (n=15). Item-level response rates were above 98% across all outcome variables.

Participants lived in Poland (n=476), the UK (n=120), Germany (n=97), Belgium/Netherlands (n=74), Croatia (n=68), Italy (n=59). The Other EU group (n=92) comprised participants from Denmark (n=39), Portugal (n=23), Spain (n=16), Ireland (n=5), Bulgaria (n=2), Albania (n=1), Cyprus (n=1), Lithuania (n=1), Norway (n=1), Romania (n=1), Sweden (n=1), Ukraine (n=1). Most respondents were mothers (92%), aged 31–40 years (71%) and in full-time or part-time employment (59%) (table 1). In terms of education, 40% of participants had received formal education up to 16–18 years or technical training, 49% had a university degree and 11% a postgraduate degree. Few participants had lived in their country of residence for <10 years (6%).

### Child characteristics
The largest CA group was CHD (n=327; 33%). Other children were diagnosed with Down syndrome (n=262; 26%), a cleft lip (n=230; 23%), spina bifida (n=112; 11%) and Down syndrome with a CHD (n=55; 6%). In terms of comorbidities, 25% of children had another CA and 43% had another health condition. The most common age category was 1–3 years (35%) and there was a slightly higher proportion of male children (56%). Just over a third of children attended school (36%), whereas 62% were not yet of school age, and 2% were either home-schooled or unable to be schooled due to their health status.

### Provision of healthcare across countries
#### Cancelled or postponed appointments
Cancellations or postponements of routine appointments were reported by 68% (623/920) of the whole sample, by 53% (427/803) for planned tests or procedures and by 26% (121/609) for planned surgeries. The UK and Poland had the largest proportions of parents reporting cancelled or postponed appointments for each category (figure 1). For routine appointments and planned tests/procedures, proportions were significantly lower in Germany, Croatia, Belgium/Netherlands and the Other EU group compared with Poland (table 2). For planned surgeries, all countries except the UK had a significantly lower proportion of cancelled or postponed appointments than Poland (full regression findings are available in online supplemental file 1).

**Table 2** Proportion of participants reporting 'cancelled or postponed' routine appointments, planned tests or procedures, and planned surgeries, by country

| Country | Routine appointments (N*=920) | | Planned tests or procedures (N*=803) | | Planned surgeries (N*=609) | |
| | Unadjusted | Adjusted† | Unadjusted | Adjusted† | Unadjusted | Adjusted† |
| | % (95% CI) | % (95% CI) | % (95% CI) | % (95% CI) | % (95% CI) | % (95% CI) |
|---|---|---|---|---|---|---|
| Poland | 79 (75 to 83) | 79 (75 to 83) | 66 (61 to 70) | 65 (60 to 70) | 37 (31 to 43) | 35 (29 to 41) |
| UK | 88 (82 to 94) | 86 (80 to 93) | 67 (58 to 76) | 67 (57 to 76) | 31 (20 to 41) | 33 (22 to 44) |
| Germany | 29 (19 to 38) | 31 (21 to 42) | 16 (8 to 24) | 18 (9 to 27) | 7 (1 to 14) | 8 (1 to 15) |
| Croatia | 46 (34 to 58) | 44 (32 to 56) | 36 (23 to 49) | 36 (23 to 49) | 14 (3 to 24) | 13 (3 to 24) |
| Italy | 69 (57 to 81) | 70 (58 to 82) | 52 (38 to 66) | 54 (40 to 68) | 9 (1 to 18) | 11 (0 to 20) |
| Belgium/Netherlands | 34 (23 to 46) | 39 (27 to 51) | 20 (9 to 31) | 23 (11 to 34) | 17 (7 to 27) | 16 (6 to 25) |
| Other EU | 56 (46 to 67) | 57 (47 to 67) | 43 (32 to 53) | 43 (32 to 53) | 12 (4 to 19) | 12 (4 to 20) |

*Total number of participants excluding 'not applicable' responses. Missing data: routine appointments (n=9), planned tests or procedures (n=8), planned surgeries (n=5).
†Adjusted by congenital anomaly type, parental age, and education level.

### Virtual appointments (by telephone or online)

Overall, 61% (544/891) of participants reported that their child's face-to-face appointments had been rescheduled as virtual appointments. This proportion was highest in the UK (87%), significantly higher than in Poland (71%). In all other countries, this proportion was significantly lower than Poland (table 3).

Overall, 29% (159/541) of participants reported their child's virtual appointments as being of 'poor' quality overall. This proportion was highest in Poland (37%) and significantly lower in the UK (21%), Belgium/Netherlands (5%) and Germany (0%) (table 3). There was a significant impact of education level on ratings, whereby more highly educated participants were less likely to rate the overall quality of their virtual appointments as 'poor' (OR =0.55, 95% CI 0.41 to 0.77; p=0.000).

### Access to medication

Overall, 26% (182/705) of participants reported some problems accessing medication for their child during the pandemic. This proportion was highest in the UK (42%) and Poland (34%) (table 3). Italy (14%), the Other EU group (8%), Germany (7%) and Croatia (4%) all had significantly fewer participants reporting problems compared with Poland (table 3).

### Impact on the child's health and well-being across countries

Overall, 30% (221/749) of participants reported that changes to their child's treatment during the pandemic had moderately to severely compromised their child's health. This figure was significantly higher in Poland (43%) compared with the UK (28%), the Other EU group

**Table 3** Proportion of participants reporting appointments rescheduled as virtual, virtual appointments rated as 'poor', and problems accessing medication, by country

| Country | Appointments rescheduled as virtual (N*=891) | | Virtual appointments rated as 'poor' (N*=552) | | Problems accessing medication (N*=713) | |
| | Unadjusted | Adjusted† | Unadjusted | Adjusted† | Unadjusted | Adjusted† |
| | % (95% CI) | % (95% CI) | % (95% CI) | % (95% CI) | % (95% CI) | % (95% CI) |
|---|---|---|---|---|---|---|
| Poland | 72 (68 to 76) | 71 (67 to 75) | 37 (32 to 42) | 37 (32 to 43) | 34 (29 to 39) | 34 (29 to 39) |
| UK | 87 (81 to 93) | 87 (81 to 93) | 21 (13 to 29) | 21 (13 to 29) | 43 (33 to 54) | 42 (32 to 52) |
| Germany | 24 (15 to 33) | 25 (16 to 35) | 0 (0 to 17) | 0 (0 to 17) | 6 (1 to 12) | 7 (1 to 14) |
| Croatia | 46 (33 to 59) | 46 (33 to 58) | 15 (2 to 29) | 17 (2 to 31) | 5 (0 to 11) | 4 (0 to 10) |
| Italy | 34 (21 to 47) | 37 (23 to 50) | 29 (8 to 51) | 27 (7 to 48) | 14 (3 to 25) | 14 (3 to 25) |
| Belgium/Netherlands | 28 (17 to 40) | 34 (22 to 46) | 6 (0 to 18) | 5 (0 to 15) | 19 (8 to 29) | 19 (8 to 29) |
| Other EU | 50 (40 to 60) | 52 (42 to 62) | 23 (10 to 35) | 22 (10 to 34) | 9 (2 to 16) | 8 (2 to 14) |

*Total number of participants excluding 'not applicable' responses. Missing data: appointments rescheduled as virtual (n=8), virtual appointments rated as 'poor' (n=11), problems accessing medication (n=8).
†Adjusted by congenital anomaly type, parental age and education level.

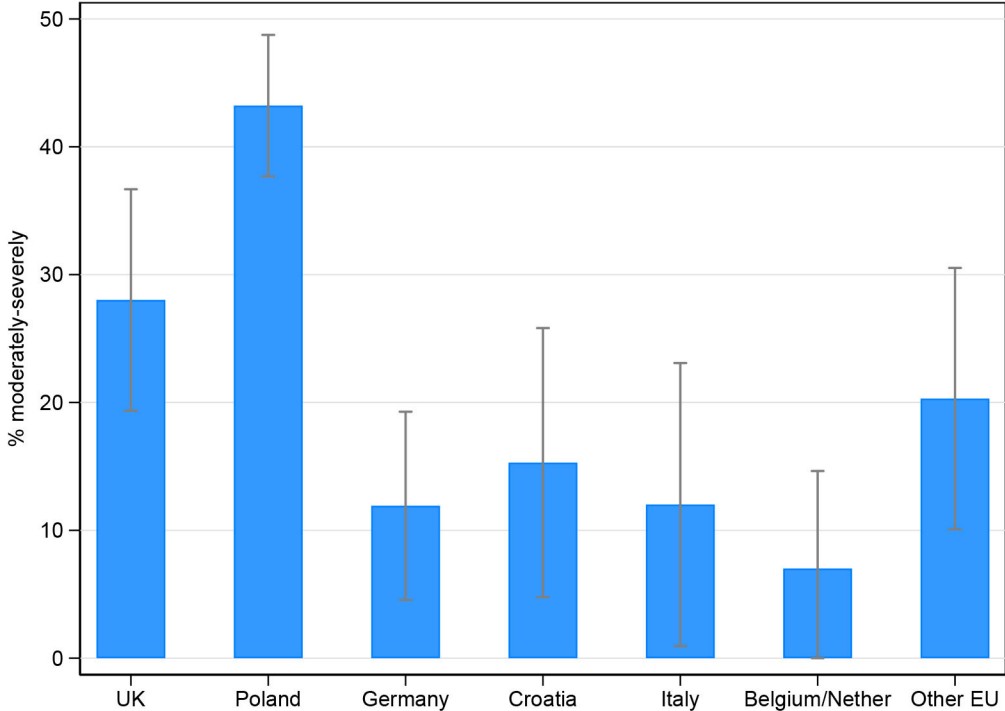

**Figure 2** Proportion* of participants reporting that their child's health had been 'moderately to severely' compromised following changes to their child's treatment with 95% CIs, by country. *Adjusted by congenital anomaly type, parental age, and education level.

(20%), Croatia (15%), Italy (12%), Germany (12%) and Belgium/Netherlands (7%) (figure 2).

The majority of participants rated their child's physical health (68%; 634/927) and emotional well-being (56%; 515/927) as being 'about the same' as it was prior to the pandemic. Overall, there was a greater proportion of participants who rated their child's emotional well-being as 'worse' (35%; 319/927) compared with 'worse' ratings for physical health (17%; 162/927).

There was a significant impact of country on ratings for physical health, with all countries less likely to rate their child's physical health as 'worse' than before the pandemic compared with Poland (figure 3). Ratings for the impact of COVID-19 on emotional well-being were similar across countries.

### Support for parents across countries

Overall, 23% (220/957) of participants reported that they would have liked more support during the pandemic 'very much'. This proportion was highest in Poland (30%) and significantly lower in Croatia (14%), Belgium/Netherlands (9%) and the other EU group (11%) (figure 4). In terms of the source of support, satisfaction ratings were lowest for support from medical sources and highest for people that participants had close relationships with, such as their partner (table 4).

### Medical sources

The UK and Poland had the lowest proportion of 'very satisfied' ratings for general practitioners (GPs), 25% and 26%, respectively (table 4). Compared with Poland,

ratings were significantly higher in Germany (85%) and the other EU group (45%). Italy and Poland had the lowest 'very satisfied' ratings for specialist doctors/nurses, 31% and 32%, respectively. Compared with Poland, ratings were significantly higher in Germany (84%) and the other EU group (45%) (table 4).

### Organisations

The highest proportion of 'very satisfied' ratings for patient organisations was for parents in Germany (59%) and Poland (56%) (table 4). Compared with Poland, these satisfaction ratings were significantly lower in the UK (38%) and Belgium/Netherlands (14%). The UK had the highest proportion of participants who were 'very satisfied' with support from their child's school (47%), however, there were no significant country-related effects.

### Close relationships

Poland had the highest proportion of 'very satisfied' ratings for support from parents of other children with the same health condition (66%), significantly higher than the UK (51%) and Belgium/Netherlands (33%) (table 4). There were no significant differences in satisfaction ratings for support from 'partner' or 'friends/family' across countries.

### Outcomes across CA types

There were few differences across CA types, with significant differences only found for items relating to the provision of healthcare. In summary, parents

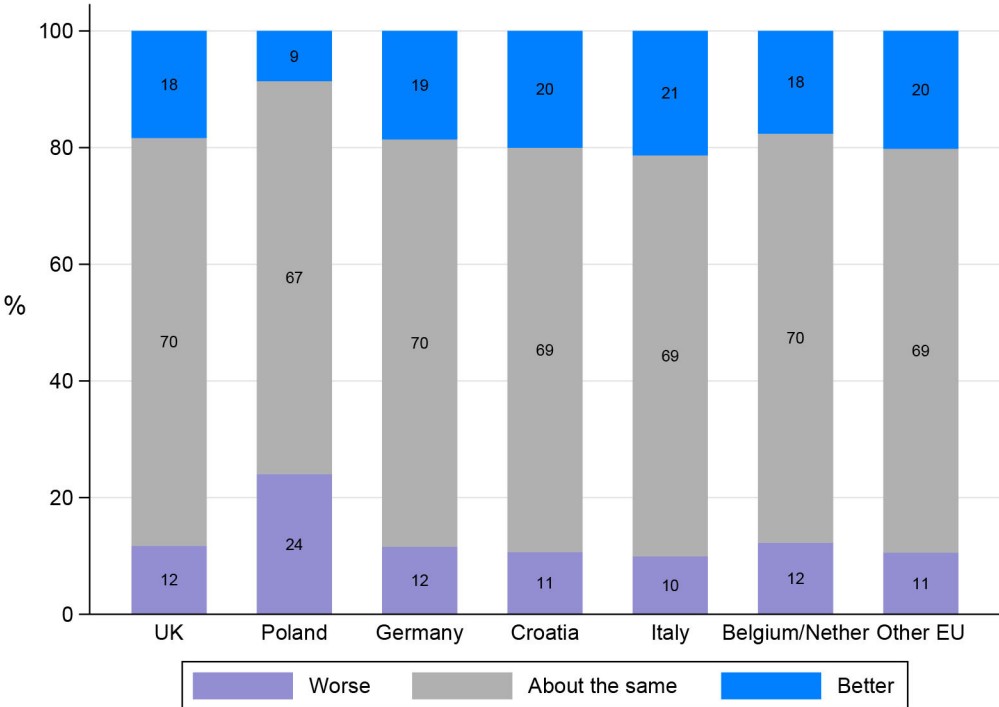

**Figure 3** Proportion* of participants reporting that their child's physical health was 'worse', 'about the same' or 'better' than it was prior to the pandemic with 95% CIs, by country. *Adjusted by congenital anomaly type, parental age, and education level.

of children with CHD (43%) reported a significantly lower proportion of 'cancelled or postponed' tests/procedures compared with parents of children with spina bifida (65%) and Down syndrome (alone) (62%) (table 5). The CHD group also reported significantly fewer rescheduled appointments (49%)

compared with the Down syndrome with CHD (80%), Down syndrome (72%) and spina bifida (70%) groups (table 5). A lower proportion of parents of children with a cleft lip (17%) reported problems accessing medication compared with the CHD group (34%) (table 5).

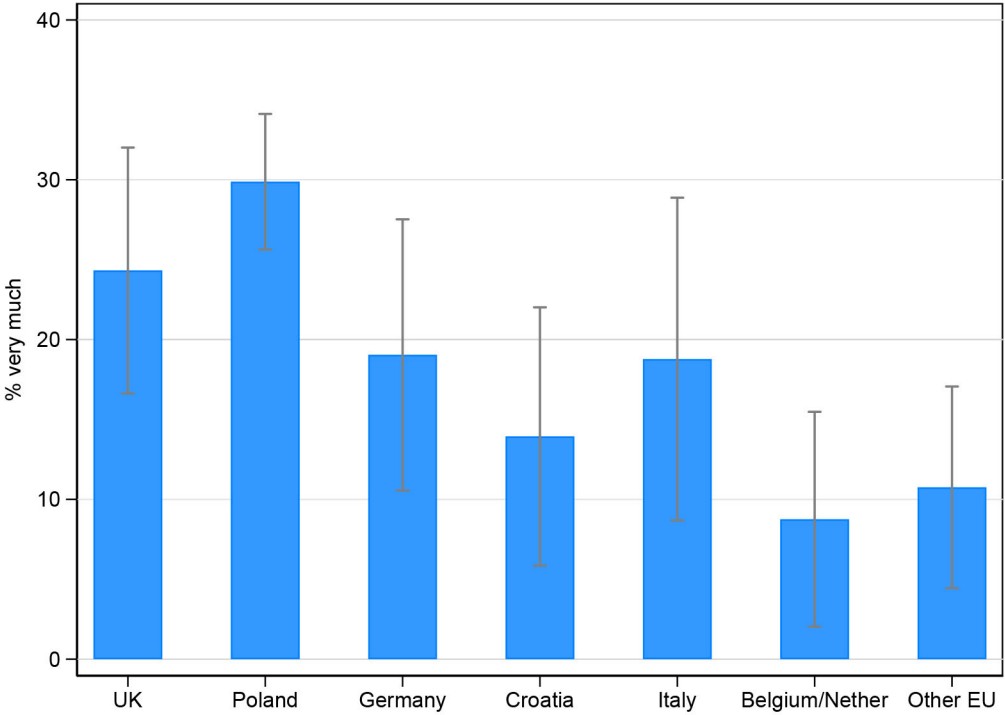

**Figure 4** Proportion* of participants reporting they would have liked more support during the pandemic 'very much' with 95% CIs, by country. *Adjusted by congenital anomaly type, parental age, and education level.

**Table 4** Proportion* of participants reporting that they were 'very satisfied' with the support they received from each source, by country

| Country | GP (N†=775) % (95% CI) | Specialist doctor/nurse (N†=792) % (95% CI) | Partner (N†=868) % (95% CI) | Friends/family (N†=883) % (95% CI) | Parents of children with same condition (N†=637) % (95% CI) | Patient organisations (N†=510) % (95% CI) | Schools (N†=369) % (95% CI) |
|---|---|---|---|---|---|---|---|
| Poland | 26 (21 to 30) | 34 (27 to 37) | 69 (65 to 74) | 61 (56 to 66) | 66 (61 to 71) | 56 (50 to 62) | 27 (20 to 34) |
| UK | 25 (16 to 34) | 39 (29 to 49) | 79 (71 to 86) | 49 (40 to 58) | 51 (41 to 61) | 38 (27 to 48) | 47 (34 to 60) |
| Germany | 85 (76 to 94) | 84 (76 to 93) | 89 (81 to 97) | 73 (62 to 85) | 60 (42 to 79) | 59 (38 to 80) | 39 (17 to 61) |
| Croatia | 41 (28 to 53) | 46 (34 to 59) | 80 (1 to 90) | 64 (52 to 75) | 64 (51 to 76) | 36 (21 to 51) | 19 (2 to 36) |
| Italy | 44 (29 to 59) | 31 (16 to 45) | 71 (58 to 84) | 47 (32 to 61) | 44 (26 to 61) | 33 (18 to 49) | 42 (27 to 57) |
| Belgium/ Netherlands | 57 (44 to 72) | 59 (46 to 71) | 66 (54 to 77) | 49 (37 to 61) | 33 (17 to 50) | 14 (0 to 27) | 36 (20 to 52) |
| Other EU | 45 (33 to 57) | 45 (34 to 56) | 70 (60 to 79) | 52 (42 to 63) | 56 (43 to 68) | 38 (26 to 51) | 33 (19 to 46) |
| Total | 37 (33 to 40) | 42 (39 to 45) | 72 (70 to 75) | 58 (55 to 61) | 60 (56 to 64) | 46 (42 to 50) | 34 (29 to 38) |

*Adjusted by congenital anomaly type, parental age and education level. Unadjusted proportions are not included in this table.
†Total number of participants completing the item, excluding 'not applicable' responses. Missing data: GP (n=10), specialist doctor/nurse (n=16), partner (n=11), friends/family (n=14), parents of children with same condition (n=17), patient organisations (n=18), schools (n=35).
GP, general practitioner.

## DISCUSSION
### Main findings
This study provides a snapshot of the healthcare experiences of children with CAs and their caregivers' experiences of support across Europe, 1 year into the COVID-19 pandemic. Overall, many participants reported disruptions to their child's routine care, which appeared to have an impact on the health of some children. Compared with non-medical organisations and parents' close relationships, parents were least satisfied with support from GPs and specialist doctors/nurses, which was particularly poor in Poland and the UK. There were also regional differences in the proportions of parents reporting disruptions to healthcare, which again appeared most severe in Poland and the UK. Few differences were found in outcomes according to CA type, suggesting that the geographical location of participants had more of an influence on healthcare experiences than the child's specific health condition.

While acknowledging that a range of factors may underpin differences across countries (such as reductions in hospital visits to minimise infections), a possible hypothesis is that these are indicative of existing vulnerabilities within local healthcare systems, with lower resourced systems being less able to meet the needs of patients during the pandemic. In relation to the healthcare workforce, figures from the Organisation for Economic Co-operation and Development in 2018 indicate that Poland had the lowest numbers of practising doctors per head in Europe (2.3/1000), closely followed by the UK (2.8/1000).[39] In contrast, Germany had one of the highest numbers per head in Europe (4.4/1000).[39] Among European countries, Poland, Italy and the UK also had below average numbers of practicing nurses per head, a factor associated with patient satisfaction with care,[40] which ranged from 5.1 to 7.8/1000.[39] In comparison, the Netherlands, Belgium and Germany all had above average figures, 11.1, 11.2 and 13.1/1000, respectively. A larger number of healthcare workers is likely to have helped with increased demand during the pandemic and helped mitigate the consequences of staff sickness.[39] Other factors that may account for regional differences include our method of recruitment and the severity of local restrictions. Participants were recruited with a consistent strategy across countries; however, some had a greater number of organsations who advertised the study and some were able to request that these organisations advertise the survey more frequently (eg, in Poland). Recruitment periods also differed, ranging from 18 weeks in the UK and Poland to only 6.5 weeks in Italy (table 1). As people are more likely to respond to surveys which are highly salient to their experiences,[41] one might expect parents experiencing more challenges during the pandemic to have been more likely to respond. However, our survey was a general survey about parents' information and support needs, with only one subsection relating to the pandemic, so this is unlikely. It is difficult to estimate the extent to which variations in the depth and duration of COVID-19 containment strategies might have influenced cross-country variations in outcomes. Containment strategies varied both regionally, within each country, and internationally, during the survey recall period (January 2020–July 2021). In addition, all countries experienced a period of full lockdown during this timeframe, making these other variations less pertinent.

**Table 5** Proportion of participants reporting 'cancelled or postponed' planned tests/procedures, appointments rescheduled as virtual, virtual appointments rated as 'poor', and problems accessing medication, by CA type

| Congenital anomaly | Planned tests or procedures (N*=803) | | Appointments rescheduled as virtual (N*=891) | | Virtual appointments rated as 'poor' (N*=552) | | Problems accessing medication (N*=713) | |
|---|---|---|---|---|---|---|---|---|
| | Unadjusted %(95% CI) | Adjusted† %(95% CI) | Unadjusted %(95% CI) | Adjusted† % (95% CI) | Unadjusted %(95% CI) | Adjusted† %(95% CI) | Unadjusted %(95% CI) | Adjusted† %(95% CI) |
| CHD | 42 (36 to 47) | 43 (37 to 49) | 46 (41 to 52) | 49 (44 to 55) | 36 (29 to 44) | 36 (28 to 44) | 31 (25 to 36) | 34 (21 to 40) |
| Cleft lip | 48 (41 to 55) | 51 (44 to 58) | 52 (45 to 58) | 56 (49 to 62) | 33 (24 to 42) | 29 (21 to 38) | 16 (9 to 22) | 17 (10 to 23) |
| Spina bifida | 64 (54 to 73) | 65 (56 to 73) | 66 (58 to 75) | 70 (62 to 77) | 30 (19 to 40) | 27 (17 to 37) | 30 (21 to 39) | 28(19 to 36) |
| Down syndrome | 67 (61 to 73) | 62 (55 to 68) | 76 (71 to 81) | 72 (67 to 78) | 28 (21 to 34) | 29 (22 to 36) | 27 (20 to 33) | 24 (18 to 30) |
| Down syndrome with CHD | 60 (47 to 74) | 55 (40 to 68) | 83 (73 to 93) | 80 (69 to 91) | 33 (18 to 47) | 28 (13 to 42) | 20 (8 to 31) | 17 (6 to 27) |

*Total number of participants excluding 'not applicable' responses. Missing data: planned tests/procedures (n=8), appointments rescheduled as virtual (n=8), virtual appointments rated as 'poor' (n=11), problems accessing medication (n=8).
†Adjusted by parental country of residence, age, and education level.
CA, congenital anomaly; CHD, congenital heart defects.

The use of virtual healthcare appointments during the pandemic meant parental concerns about cancellations could be addressed, while limiting their exposure to SARS-CoV-2.[19] Around 60% of our sample reported that they had a face-to-face appointment rescheduled as virtual. This was higher than other in other studies conducted with paediatric patients, which found that 11%[18] and 20%[23] of participants reporting rescheduled appointments. However, these studies were conducted during the first wave of the pandemic (April–May 2020), a year before our survey was delivered, so this rise is not unexpected. The quality of virtual appointments was rated as 'fair–excellent' by two-thirds of our sample, whereas one-third rated them as 'poor'. This is in line with findings from a similar survey, which found that 68% of CHD parents and patients described their virtual appointments as 'adequate'.[23] In contrast, other studies conducted within specialist paediatric centres found higher satisfaction ratings,[19 42] such as 87% mean satisfaction ratings for virtual paediatric appointments.[43] This increase, however, may be due to a slight social-desirability bias as parents were asked to rate satisfaction by staff within the specialist centres, whereas in our survey, the recruiters were not involved in the child's care. Of note is our finding whereby participants with a lower level of education were more likely to rate virtual appointments as poor.

### Strengths and limitations
This study surveyed parents and carers of children with different CA types within several countries, including a broad range of experiences. We recruited a large sample of parents and carers overall, and the proportion of each CA type reflects the relative number of live births with each CA in Europe.[44] Although the survey was shared widely, the use of convenience sampling means that there is a risk of selection bias, and the views and experiences of this sample may not be representative of all parents and carers of children with CAs. The use of social media to recruit participants may have excluded people living with 'digital poverty' and people who do not tend to engage with these types of organisations, whose experiences may differ from this sample. When considering recruitment figures within each country, these were mostly small. The survey was developed with input from parents of children with a CA, however, we were unable to conduct a full pilot of the final version. Great care was taken to avoid leading questions, ambiguity or complex language; however, it is possible that there may have been some issues with the wording or content of survey items.

### Implications and future research
Our survey findings are important and provide useful insights into the provision of care for children with CAs across Europe during the first year of the pandemic. Findings highlight potential weaknesses of healthcare systems in some countries and suggest that long-term systemic action is required to improve patient experiences and outcomes. The situation appears particularly problematic

in the UK and Poland, which may benefit from increased resources to provide for this vulnerable group of patients. Patient organisations and charities provide an invaluable source of knowledge and support to parents of children with CAs, and these should be supported, especially in countries where medical capacity to meet patients' needs may be stretched.

As with many other patient groups, it is clear that the COVID-19 pandemic has had an impact on the experiences of children living with a serious health condition and their families.[20 22 45] This particular survey suggests disruptions to care for children, with potential impacts on the child's health and well-being. Considering the limitations of this study, it will be important to further investigate the impact of the COVID-19 pandemic on the delivery of paediatric services across Europe using population-based data. With the proliferation of telemedicine to deliver care during the pandemic,[43] exploring the reasons why these virtual strategies were lacking for some parents (particularly those with a lower level of education) is important to ensure optimal parental satisfaction with future care and support from medical professionals.

## Conclusion

The COVID-19 pandemic continues to put pressure on healthcare systems worldwide. Our survey findings highlight disruptions to the delivery of care across Europe, particularly in the UK and Poland, which raises questions about the ability of the healthcare systems within these countries to meet the needs of children with CAs and their families, and a need for increased resources.

**Author affiliations**

[1]Chair and Department of Medical Genetics, University of Medical Sciences, Poznan, Poland
[2]Population Health Research Institute, St George's University of London, London, UK
[3]Population Health Sciences Institute, Faculty of Medical Sciences, Newcastle University, Newcastle, UK
[4]Children's Hospital Zagreb, Centre of Excellence for Reproductive and Regenerative Medicine, Medical School University of Zagreb, Zagreb, Croatia
[5]Rare Diseases Research Unit, Fundacio per al Foment de la Investigacio Sanitaria i Biomedica, Valencia, Spain
[6]Provincial Institute for Hygiene, Antwerpen, Belgium
[7]Department of Paediatrics and Adolescent Medicine, Lillebaelt Hospital, University Hospital of Southern Denmark, Kolding, Denmark
[8]Department of Epidemiology, National Health Institute Doutor Ricardo Jorge, Lisboa, Portugal
[9]Department of Genetics, University Medical Center Groningen, Groningen, The Netherlands
[10]Department of Epidemiology, National Institute of Health, Lisbon, Portugal
[11]IMER Registry (Emilia Romagna Registry of Birth Defects), University of Ferrara and Azienda Ospedaliero Universitaria di Ferrara, Ferrara, Italy
[12]Malformation Monitoring Centre Saxony-Anhalt, Medical Faculty Otto-von-Guericke University, Magdeburg, Germany
[13]Unit of Epidemiology of Rare Diseases and Congenital Anomalies, Institute of Clinical Physiology, Pisa, Italy
[14]Population Health Research Institute, St George's, University of London, London, UK

**Acknowledgements** The authors are hugely grateful to all the parents and carers who took part in the study. We thank the following people for their support in developing the survey and its dissemination in Poland: Dominika Madaj-Solberg (Spina Foundation, Katowice, Poland), Tomek i Kasia Grybek (Borys the Hero Foundation, Gdańsk, Poland), Halina Grzymisławska-Słowińska (Fundacja TAK dla Samodzielności, Poznań, Poland), Anna Latos (Bydgoszcz, Poland), Professor Jolanta Wierzba (Med.Univ. Gdańsk, Poland), Professor Robert Śmigiel (Med. Univ. Wrocław, Poland), Professor Olga Haus (Coll.Med. UMK, Bydgoszcz, Poland), and Dorota Trześniewska (Poznań, Poland). We thank the following people and organisations for advertising the survey across Europe: The Cleft Lip and Palate Association, The Children's Heart Federation, International Federation for Spina Bifida and Hydrocephalus, Children's Heartbeat Trust, Down's Syndrome Association, and Down Syndrome International, Dr Nadia Assanta (Fondazione Toscana Gabriele Monasterio), Dr Giada Cavazzuti (Associazione 'Un cuore, un mondo'), Dr Elisabetta Lapi, Dr Antonella Falugiani (Associazione 'Trisomia 21 Onlus'), Dr Alessandro Giacomina, Dr Marina Rossi (AOU Pisana), Jürgen Wolters (Arbeitsgemeinschaft Spina Bifida und Hydrocephalus, ASBH), Dr Annett Lambrecht (Department of Pediatric Cardiology, University Hospital Magdeburg), Dr Christian Zahl (Department of Oral and Maxillofacial Surgery, University Hospital Magdeburg), Hjerteforeningens børneklub (Kid's Heart Association Club), Rygmarvsbrokforeningen (Spina Bifida and Hydrocephalus Association), Downs syndrom Danmark (Danish Down Syndrome), Landsforeningen Læbe- Ganespalte (Cleft Lip and Palate Association), Pais21 (Down Portugal/ Down Syndrom Parents Association Pais 21), Associação Spina Bifida e Hidrocefalia de Portugal (Spina Bifida and Hidrocephalus Portugal Association), Associação Coração Feliz (Happy Heart Association), Associação Portuguesa dos Amigos das Crianças Portadoras de Fendas Lábio-Palatinas (Portugese Association of Children with Lip-Palatine Clefts), The Foundation for the Promotion of Health and Biomedical Research of Valencia Region (FISABIO), Universitair Ziekenhuis Antwerpen (University Hospital of Antwerp), Vereniging voor Aangeboren Gelaatsafwijkingen (Association for Congenital Facial Defects), Dr Annick Laridon - Het Centrum voor Ontwikkelingsstoornissen (Centre for Developmental Disorders), Mario Sel - Spina Bifida Hydrocephalus Belgium, Hrvatski savez za rijetke bolesti (Rare Diseases Croatia), Veliko srce malom srcu (A Big Heart For Little Heart), Hrvatska zajednica za Down sindrom (Croatian Down Syndrome Association), Udruga roditelja djece s rascjepom usne i/ili nepca OSMIJEH (Association of Children With Cleft Lip With/Without Cleft Palate), Udruga Aurora- Udruga roditelja i djece sa spinom bifidom (Aurora Association- Association of Parents and Children with Spina Bifida), Patientenvereniging Aangeboren Hartaandoeningen (Congenital Heart Disease Association), De 'Stichting Downsyndroom' (Down Syndrome Association). We are very grateful to Esben Garne Holm and Juan Rico for supporting the translation of the survey.

**Contributors** AL-B conceptualised the study. AL-B, EM, JKM, and JR contributed to the study design and survey development. EM led the survey development, translation, and recruitment of participants. EM and JKM conducted the data analysis. AL-B, JKM, and JR critically revised the manuscript. AJ-D, IB, CC-C, EDH, EG, EM, LG, AJS, LRL, CMD, CNP, AJN, AN, LO, LP-R, AP and AR oversaw the translation of the survey and recruited participants. EM drafted the manuscript. All authors contributed to, read and approved the final manuscript. JKM acts as a guarantor for the manuscript.

**Funding** This project has received funding from the European Union's Horizon 2020 research and innovation programme under grant agreement number 733001. Start date: 1 January 2017. Duration: 5 years and 5 months. The views presented here are those of the authors only, and the European Commission is not responsible for any use that may be made of the information presented here.

**Competing interests** None declared.

**Patient and public involvement** Patients and/or the public were involved in the design, or conduct, or reporting, or dissemination plans of this research. Refer to the Methods section for further details.

**Patient consent for publication** Not applicable.

**Ethics approval** Ethics approval for the study was granted by the St George's (University of London). Research Ethics Committee on 18 December 2020 (reference number: 2020.0311). In Poland, ethics approval was granted on 10 December 2020 by the Bioethics Committee at the Poznań University of Medical Sciences (reference number: 882/20). In Croatia, ethics approval was granted on 10 December 2020 by the Ethics Committee of the Children's Hospital Zagreb (Protocol No: 02-23/43-1-20 Zagreb). In Spain, ethics approval was granted on 21 December 2020 by the Clinical Investigation Ethics Committee of the "Dirección General de Salud Pública y Centro Superior de Investigación en Salud Pública" (reference number: 20201221/05). In Belgium, ethics approval was granted on 1 March 2021 by the Ethics Committee of the University Hospital of Antwerp (reference: 21/06/084). In Portugal, ethics approval was granted on 16 March by the Ethics Committee of the National Institute of Health Doutor Ricardo Jorge (CES-INSA). In

Germany, ethics approval was granted on 15 April 2021 by the Medical Faculty of the Otto-von-Guericke-University Magdeburg Research Ethics Committee (reference number: 44/21). In Italy, ethics approval was granted on 14 June 2021 by the Research Ethics and Integrity Committee of the National Research Council Institute of Clinical Physiology in Pisa (CNR-INF) (protocol number 0065527/2019). No local ethics approvals were required in Denmark (Lillebaelt Hospital—University Hospital of Southern Denmark) or the Netherlands (University Medical Center Groningen).

**Provenance and peer review** Not commissioned; externally peer reviewed.

**Data availability statement** The datasets analysed during the current study are available from the corresponding author on reasonable request.

**ORCID iDs**
Elena Marcus http://orcid.org/0000-0003-0900-8976
Clara Cavero- Carbonell http://orcid.org/0000-0002-4858-6456
Ester Garne http://orcid.org/0000-0003-0430-2594
Lucía Páramo-Rodríguez http://orcid.org/0000-0002-2952-564X
Anke Rissmann http://orcid.org/0000-0002-9437-2790
Joan K Morris http://orcid.org/0000-0002-7164-612X

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
