## [Reviewer comments · BMJ Open]

ARTICLE DETAILS

TITLE (PROVISIONAL)	COVID-19 and children with congenital anomalies: a European survey of parents' experiences of healthcare services.
AUTHORS	Latos-Bieleńska, Anna; Marcus, Elena; Jamry-Dziurła, Anna; Rankin, Judith; Barisic, Ingeborg; Caverio- Carbonell, Clara; Den Hond, Elly; Garne, Ester; Genard, Lucas; Santos, Ana; Lutke, Renee; Dias, Carlos; Neergaard Pedersen, Christina; Neville, Amanda; Niemann, Annika; Odak, Ljubica; Páramo-Rodríguez, Lucía; Pierini, Anna; Rissmann, Anke; Morris, Joan

VERSION 1 – REVIEW

REVIEWER	Bay, Annika Umeå Universitet Medicinska fakulteten, Nursing
REVIEW RETURNED	18-Feb-2022

GENERAL COMMENTS	Covid-19 and children with congenital anomalies: a European survey of parents' experiences of healthcare services. Overall the study presents very interesting and important findings. The Covid-19 pandemic will have many consequences regarding follow-ups, and this case study fills a gap concerning lack of follow-ups. The paper is well written and nice to read. The method section and discussion sections needs extra attention, I lack a more detailed description overview from the analysis procedure. Although I like the Figure, I miss the detailed text that explains how you concluded the findings. There are very few references, and a lot of them are from your own earlier research. Please include other research as well, it will strengthen your article. Titel: Page 3, line 3; Should be consistent with the aim of the study, in the title it is the parents' experiences and in the aim the children? Abstract Objective: To survey parents and carers of children with a congenital anomaly (CA) across Europe about their child's experiences of healthcare services, and their experiences of support during the COVID-19 pandemic. I can not find anything in your article about the child's experiences? Skip "the child's" Background Page 7, line 9; It will be a misdirection, it is the parents experiences and not the children's. On the same page line number 19, you have another aim that is more in line with what you have studied. Methods Recruitment
---

	Page 9, line 22; Was there any difference between the countries in how the information about the study was communicated? Was there any way that excelled, such as social media? Could it be a bias that perhaps it was the parents who were dissatisfied with the follow-up who were the ones who responded to the survey? It may be a limitation. Results It is sometimes not easy to follow your results in the tables, maybe you can elaborate them more? Since you used logistic regression, it would be desirable for you to report OR and p-value in your tables. CI should be reported as an interval, for example 41-55. Discussion The discussion seems to primarily repeat your results, it would be a strength to discuss more clearly how these findings relate to other relevant literature on follow-ups. You have some references but I think it could elevate your article if you included other people's perspectives. In this section, I miss discussion about the fact that Italy came in very late in the process, could it have any significance for their low results? Their health care was also hit hard by the pandemic early. Page 19, line 51; Due to your possible hypothesis, it would have been interesting to know how the parents experiences the care before the pandemic. Figure Please name the figures (The text that you have under Figures legend).
--	---

REVIEWER	Wray, Jo Great Ormond Street Hospital for Children, Heart and Lung Directorate
REVIEW RETURNED	04-Mar-2022

GENERAL COMMENTS	This is a well written, clearly presented manuscript on an important topic. To enable the reader to fully understand the findings the questionnaire should be made available as supplementary material or as part of the main article. I could not see any supplementary material - was the STROBE checklist submitted with the manuscript? Some specific points - the authors might wish to look at the manuscript of Cousino and colleagues which also described an international survey on the impact of COVID-19 on the delivery of health care services for the CHD population (adults and children) but also included measures of psychological stress (Cousino MK et al. Impact of the COVID-19 pandemic on CHD care and emotional wellbeing. Cardiol Young. 2021 May;31(5):822-828. doi: 10.1017/S1047951120004758. PMID: 33308334). How were potential participants given the link to the questionnaire - was this via the social media sites? Is there any further information about the numbers who had to complete a consent form and where they were from? Did parents have the option to state that they chose to cancel the face to face appointment?
---

	How did cancellation rates for surgery and other planned procedures compare with usual cancellation rates? Were parents just asked whether appointments, procedures had been cancelled or were any questions asked about rescheduling, for example? Was any further detail asked about reasons for the virtual appointments being 'poor' - e.g. was it related to technology, scheduling, the interaction with the staff etc. In relation to support, were specialist doctors/nurses all included as one question? If so, I suggest this is mentioned in the limitations. How was 'support' defined? Although not the focus of this study, some discussion about the positive findings in terms of families who reported that their child's physical health was better than pre-COVID is warranted - this could also be linked to health service delivery, for example more tests being undertaken locally with greater attention to the results - as well as to factors such as lockdown and the child being less likely to get infections etc (Stewart et al, 2021). The association between education level and rating of virtual clinics warrants further discussion, in terms of how parents with lower levels of education could be supported better to access and engage with virtual appointments. Whilst the findings are important and do provide some insights about the provision of health care to children in different countries, there are a number of limitations which should be more clearly articulated in relation to recruitment methods, access to the survey and any potential biases, the use of a survey to collect these data and the lack of depth to the responses, the impact of other factors such as parental anxiety, other COVID-related disruptions, cultural differences, other life experiences which might have affected parental responses, inability to calculate a response rate or generalise the findings etc. In the discussion please clarify how the survey has helped to identify best practices that may help address current challenges, given this was one of the aims. What were the best practices? The authors discuss the fact that some countries may benefit from increased resources to provide care for children with congenital anomalies - what resources? To what extent were some of the care delivery issues about resources rather than pragmatic decisions to reduce risk? Further discussion would be helpful, considering also the context of the different systems for delivering health care. In summary, this paper provides important, albeit superficial, data about country differences in relation to the delivery of health care to children during the pandemic and has highlighted potential vulnerabilities of some health care systems. Further detail, particularly in relation to the questionnaire, would help the reader to interpret the findings.
--	---

REVIEWER	Aboulatta, Laila University of Manitoba, College of Pharmacy
REVIEW RETURNED	16-Mar-2022

GENERAL COMMENTS	applaud the authors for their work and have some comments to improve their manuscript:  1. Briefly explain the context in which every country had different restrictions taken and the degree of restrictions to healthcare may have influenced the health care access and quality of children care with different degrees. Also, the duration of the pandemic lockdown across countries are likely to have affected the results especially that the examined period in Italy was from 16 June to 31 July. The authors need to explain how the that short pandemic period examined can be compared with other countries. 2. It would help if the discussion offers comparisons with similar studies and highlighting the conflicting/similar results with studies conducted in different countries. What would the authors want the readers to take home from the manuscript compared to other studies? 3. In the method section, the authors mentioned that survey was conducted as part of a collaborative European to Establishing a linked European Cohort of Children with CAs (EUROlinkCAT). Please add reference to EUROlinkCAT. 4. The authors mentioned that the survey included the following sections: (1) Parent Demographics (9 items), (2) Child Demographics and Medical Information (7 items), (3) Provision of Healthcare (7 items), (4) Impact on the Child (4 items), and (5) Support for Parents (2 items). Please add the survey with the examined items as an appendix. Also, the authors reported that study was reported following STROBE guidelines. Please add the strobe to the appendix. 5. Data collection section: Please clarify if there is a standard for Procedure (SOP) for the question to be included in the study regarding the missing information since a standardized questionnaire is needed for skipped questions to reduce respondent bias. Also, which questions were considered critical other than country and CA. 6. Since this is an online survey only, the authors should mention that the results are not generalizable as a limitation.
---

VERSION 1 – AUTHOR RESPONSE

Reviewer Comments

Reviewer: 1

Dr. Annika Bay, Umeå Universitet Medicinska fakulteten

Comments to the Author: Covid-19 and children with congenital anomalies: a European survey of parents' experiences of healthcare services.

1. Overall the study presents very interesting and important findings. The Covid-19 pandemic will have many consequences regarding follow-ups, and this case study fills a gap concerning lack of follow-ups. The paper is well written and nice to read.

We thank the reviewer for these comments.

2. The method section and discussion sections needs extra attention, I lack a more detailed description overview from the analysis procedure. Although I like the Figure, I miss the detailed text that explains how you concluded the findings.

Thank you for these comments, we have addressed these points below, individually.

3. There are very few references, and a lot of them are from your own earlier research. Please include other research as well, it will strengthen your article.

Thank you for highlighting this. We have added additional references to the background and discussion sections.

4.

Title:

Page 3, line 3; Should be consistent with the aim of the study, in the title it is the parents' experiences and in the aim the children?

Abstract

Objective: To survey parents and carers of children with a congenital anomaly (CA) across Europe about their child's experiences of healthcare services, and their experiences of support during the COVID-19 pandemic. I can not find anything in your article about the child's experiences? Skip "the child's"

Background

Page 7, line 9; It will be a misdirection, it is the parents experiences and not the children's. On the same page line number 19, you have another aim that is more in line with what you have studied.

Thank you for highlighting this. We have amended the abstract, background and aim of the study so that it is clear that it refers to parents' experiences of healthcare services as opposed to the child's (Page 5, line 7; page 7, line 12).

5. Methods

Recruitment

Page 9, line 22; Was there any difference between the countries in how the information about the study was communicated? Was there any way that excelled, such as social media?

Could it be a bias that perhaps it was the parents who were dissatisfied with the follow-up who were the ones who responded to the survey? It may be a limitation.

Thank you for this observation. Information was communicated online via social media and relevant organisations across all countries. There were differences, however, in the number of organisations able to support recruitment within each country, as well as the frequency with which they were able to post study adverts. The best responses were received from adverts posted by patient and parent organisations which had a large following. Study adverts posted by our team on Twitter and Facebook were less successful. We agree there is a risk of selection bias, however, the survey was a general survey about the information and support needs of parents, with COVID-19 included only as a subsection. We therefore think it is unlikely that there was a specific bias relating to parents' experiences during the pandemic. We have elaborated on these points in the discussion section of the manuscript (page 19, line 54).

6. Results

It is sometimes not easy to follow your results in the tables, maybe you can elaborate them more? Since you used logistic regression, it would be desirable for you to report OR and p-value in your tables. CI should be reported as an interval, for example 41-55.

Many thanks for this observation. We have amended the CIs so that these are more clearly identifiable as an interval (Tables 2-5). We have also included the full regression findings (including OR and p-values) in a supplementary file. We feel that the adjusted proportions resulting from the logistic regressions are the most informative information for the reader.

7. Discussion

The discussion seems to primarily repeat your results, it would be a strength to discuss more clearly how these findings relate to other relevant literature on follow-ups. You have some references but I think it could elevate your article if you included other people's perspectives.

Thank you for highlighting this. We have referenced more studies in the discussion section to contextualise our findings (pages 19-20).

8. In this section, I miss discussion about the fact that Italy came in very late in the process, could it have any significance for their low results? Their health care was also hit hard by the pandemic early.

Thank you for this point. It is possible that Italy coming in very late in the process may have influenced their results, however (as discussed above), this is unlikely to be a large source of bias because the survey was a general survey about the information and support needs of parents, with only one subsection covering the pandemic. We have added this point to the discussion section (page 19, line 54).

9. Page 19, line 51; Due to your possible hypothesis, it would have been interesting to know how the parents experiences the care before the pandemic.

Thank you. We agree it is would be very interesting to know about parents' experiences prior to the pandemic. There is a substantial body of evidence, which suggests there were large reductions in pediatric healthcare visits during the pandemic compared to pre-pandemic averages. We have added this to our background section (page 7, line 3).

10. Figure

Please name the figures (The text that you have under Figures legend).

Thank you for highlighting this. We are unsure why the figures do not have a title. We have amended this in this re-submission.

Reviewer: 2

Dr. Jo Wray, Great Ormond Street Hospital for Children

Comments to the Author:

1. This is a well written, clearly presented manuscript on an important topic.

We thank the reviewer for these comments.

2. To enable the reader to fully understand the findings the questionnaire should be made available as supplementary material or as part of the main article.

Thank you, we have included the questionnaire items within the supplementary file.

3. I could not see any supplementary material - was the STROBE checklist submitted with the manuscript?

We did submit the STROBE checklist, but it seems this was not received. We have attached it again in this re-submission within the supplementary file.

4. Some specific points - the authors might wish to look at the manuscript of Cousino and colleagues which also described an international survey on the impact of COVID-19 on the delivery of health care services for the CHD population (adults and children) but also included measures of psychological stress (Cousino MK et al. Impact of the COVID-19 pandemic on CHD care and emotional wellbeing. *Cardiol Young*. 2021 May;31(5):822-828. doi: 10.1017/S1047951120004758. PMID: 33308334).

Thank you for sharing this relevant study. We did not come across this study during our searches and have added it to our manuscript.

5. How were potential participants given the link to the questionnaire - was this via the social media sites? Is there any further information about the numbers who had to complete a consent form and where they were from?

Thank you for these questions, we have added some further information about this to the paper. Potential participants were given a link to the survey website which included each language version of the survey. This was done via our own EUROLINKCAT twitter account, and the social media sites (Twitter and Facebook) of relevant charities and organisations. For example, in the UK the following organisations posted study adverts linking potential participants to our website on Facebook and Twitter, or re-tweeted our EUROLINKCAT advert: The Cleft Lip and Palate Association, The Children's Heart Federation, International Federation for Spina Bifida and Hydrocephalus, Children's Heartbeat Trust, Down's Syndrome Association, and Down Syndrome International. We also contacted relevant closed support groups on Facebook and asked permission to post information about the study within these smaller online groups.

The following countries required participants to complete an online consent form: UK, Belgium, Netherlands, Germany, Spain, Portugal, Italy. In Poland, Croatia, and Denmark consent was implied by completion of the survey.

We are not able to estimate how many potential participants were reached to calculate response rates, however, we have calculated 'submission rates', i.e. the number of participants who started the survey/number who completed and submitted the survey. We have added information about the 'submission rate' to the manuscript (page 12, line 9, 21).

For completeness, the rates in each language version of the survey are shown in the table below. Please note these figures do not match the figures in the manuscript as participants completing each language version did not necessarily reside in that particular country, and this table does not account for missing country and CA data.

Language version	Completed	Incomplete	Total	Submission rate
Poland	536	105	641	84%
UK	153	31	184	83%
Belgium	55	5	60	92%
Netherlands	26	6	32	81%
Croatia	78	19	97	80%
Germany	97	8	105	92%
Italy	61	17	78	78%
Portugal	23	4	27	85%
Spain	23	5	28	82%
Denmark	38	8	46	83%

6. Did parents have the option to state that they chose to cancel the face to face appointment? How did cancellation rates for surgery and other planned procedures compare with usual cancellation rates? Were parents just asked whether appointments, procedures had been cancelled or were any questions asked about rescheduling, for example? Was any further detail asked about reasons for the virtual appointments being 'poor' - e.g. was it related to technology, scheduling, the interaction with the staff etc.

Thank you for raising these questions. We did not include an item or a response option which allowed parents to state whether they chose to cancel face-to-face appointments themselves. We did not collect any data which allows us to compare these cancellation rates with pre-pandemic averages. We did ask parents whether they had any face-to-face appointments re-scheduled as virtual (telephone/online), of which 61% responded that they had (page 13, line 46). In terms of re-scheduled appointments, we did not ask what kind of appointments these were. In relation to the quality of virtual appointments, we asked parents to give an overall assessment of their experience (Poor, Fair, Good, or Very good). We have discussed our findings relating to face-to-face/virtual appointments in more detail in the discussion section (page 20, line 14).

7. In relation to support, were specialist doctors/nurses all included as one question? If so, I suggest this is mentioned in the limitations. How was 'support' defined?

Yes, we had a single question which asked about parents' satisfaction with support from specialist doctors and specialist nurses. We included this as a single question because we were interested in the comparison between specialists and generalists (i.e. general practitioners), and therefore do not feel that this is an important limitation to discuss. We did not define the term 'support' as we felt it was most important to understand the extent to which parents "felt satisfied" with support, as opposed to the extent to which different types of support had been delivered and received by parents. We have provided all survey items in a supplementary file so the readers can review the wording of items and any definitions.

8. Although not the focus of this study, some discussion about the positive findings in terms of families who reported that their child's physical health was better than pre-COVID is warranted - this could also be linked to health service delivery, for example more tests being undertaken locally with greater attention to the results - as well as to factors such as lockdown and the child being less likely to get infections etc (Stewart et al, 2021).

Thank you for highlighting this finding and sharing this study. As the reviewer notes this was not the focus of the study, and due to word limitations, we have not included any additional text to discuss this.

9. The association between education level and rating of virtual clinics warrants further discussion, in terms of how parents with lower levels of education could be supported better to access and engage with virtual appointments.

We agree this is an important area for future research and have added some text in the discussion (page 20, line 36; page 21, line 8).

10. Whilst the findings are important and do provide some insights about the provision of health care to children in different countries, there are a number of limitations which should be more clearly articulated in relation to recruitment methods, access to the survey and any potential biases, the use of a survey to collect these data and the lack of depth to the responses, the impact of other factors such as parental anxiety, other COVID-related disruptions, cultural differences, other life experiences which might have affected parental responses, inability to calculate a response rate or generalise the findings etc.

Thank you for highlighting these points. We were asked to provide strengths and limitations of the study as bullet points and were thus limited in the extent to which we could discuss these points. We have now added more discussion about the study limitations in the main body of the manuscript (page 19, line 46; page 20, line 40).

11. In the discussion please clarify how the survey has helped to identify best practices that may help address current challenges, given this was one of the aims. What were the best practices?

Thank you for raising this. The aim of identifying 'best practices' related to the information and support items of our survey (not covered in this paper). We have removed this aim from this paper, as the COVID-19 related items do not include a sufficient level of detail to address this.

12. The authors discuss the fact that some countries may benefit from increased resources to provide care for children with congenital anomalies - what resources? To what extent were some of the care delivery issues about resources rather than pragmatic decisions to reduce risk? Further discussion would be helpful, considering also the context of the different systems for delivering health care.

Thank you for this comment. In terms of 'increased resources' we refer to the number of practicing nurses and doctors within each country. We agree that some of the care delivery issues may also be about reducing risk of infection and have touched on this in our background section (page 6, line 35 6) and discussion section (page 19, line 26).

13. In summary, this paper provides important, albeit superficial, data about country differences in relation to the delivery of health care to children during the pandemic and has highlighted potential vulnerabilities of some health care systems. Further detail, particularly in relation to the questionnaire, would help the reader to interpret the findings.

Thank you, we agree with this point and have made the limitations to our research clearer in the discussion.

Reviewer: 3

Miss Laila Aboulatta, University of Manitoba

Comments to the Author:

Applaud the authors for their work and have some comments to improve their manuscript:

1. Briefly explain the context in which every country had different restrictions taken and the degree of restrictions to healthcare may have influenced the health care access and quality of children care with different degrees. Also, the duration of the pandemic lockdown across countries are likely to have affected the results especially that the examined period in Italy was from 16 June to 31 July. The authors need to explain how the that short pandemic period examined can be compared with other countries.

We are sorry that it was not clear that the dates 16 June to 31 July 2021 for Italy refer to the recruitment time period and not to the time frame of the survey questions. We have redrafted this for clarity (page 8, line 34; page 10, line 3). The parents were asked to reflect on their experiences during the pandemic from January 2020 until the time at which they completed the survey. We agree the degree of restrictions did vary by country, but all countries did experience a time of total lock down reducing other differences to being minor issues. We discuss this in the discussion section (page 20, line 3).

2. It would help if the discussion offers comparisons with similar studies and highlighting the conflicting/similar results with studies conducted in different countries. What would the authors want the readers to take home from the manuscript compared to other studies?

Thank you for this point. We have referenced additional studies in the discussion section to contextualise our findings.

3. In the method section, the authors mentioned that survey was conducted as part of a collaborative European to Establishing a linked European Cohort of Children with CAs (EUROlinkCAT). Please add reference to EUROlinkCAT.

Many thanks highlighting this omission. We have added a reference for EUROlinkCAT on page 7, line 16.

4. The authors mentioned that the survey included the following sections: (1) Parent Demographics (9 items), (2) Child Demographics and Medical Information (7 items), (3) Provision of Healthcare (7 items), (4) Impact on the Child (4 items), and (5) Support for Parents (2 items). Please add the survey with the examined items as an appendix. Also, the authors reported that study was reported following STROBE guidelines. Please add the strobe to the appendix.

Thank you for highlighting this. The STROBE checklist was submitted but seems it wasn't attached to the manuscript. We have uploaded this again, along with the survey items, within the supplementary file.

5. Data collection section: Please clarify if there is a standard for Procedure (SOP) for the question to be included in the study regarding the missing information since a standardized questionnaire is needed for skipped questions to reduce respondent bias. Also, which questions were considered critical other than country and CA.

Thank you for this comment. We did not use an SOP to determine which questions should be made mandatory. The only items which were considered critical were country and CA type as these were our main inclusion criteria and focus of the analysis. For all other items missing data were very low (see page 12, line 28).

6. Since this is an online survey only, the authors should mention that the results are not generalizable as a limitation.

Thank you for this comment. We touched on this limitation in the strengths and limitations section of the manuscript, and have now described this more fully in the discussion section (page 20, line 40).

VERSION 2 – REVIEW

REVIEWER	Bay, Annika Umeå Universitet Medicinska fakulteten, Nursing
REVIEW RETURNED	13-May-2022
GENERAL COMMENTS	I have had my questions clarified and they have also been addressed. Looking forward to reading the finished article.